# Strategic Charitable Giving and R&D Innovation of High-Tech Enterprises: A Dynamic Perspective Based on the Corporate Life Cycle

**Zhengwen Lu [1], Yujie Zhang [1,\*] and Yuanxu Li [2]**

1   School of Economic and Management, Shanghai Institute of Technology, Xuhui District, Shanghai 200235, China
2   School of Management, Fudan University, Handan District, Shanghai 200433, China
\*   Correspondence: zyjwzw2017@163.com; Tel.: +86-187-0340-8854

**Abstract:** As important components of differentiation strategy, charitable giving and R&D innovation can have a profound impact on the survival and growth of high-tech enterprises. However, the strategic interaction between them has not been studied in depth using the whole-life-cycle perspective. With Chinese A-share-listed high-tech enterprises in the 2015–2020 period as the research sample, the Tobit model was used to empirically test the interaction between charitable giving and R&D innovation and analyze differences in their utility over different life cycles. The results show that there was a strategic synergy between charitable giving and R&D innovation and charitable giving could significantly improve R&D innovation, but its utility was affected by changes in the life cycle of firms. Among them, the synergy utility was highest for maturing firms, followed by declining firms, but not significant for growing firms. A further study on the synergistic utility of mature firms found that, for non-state firms where executives have an R&D background, charitable giving could promote integration of external advantageous resources and R&D innovation development. Finally, the regression findings remained significant after accounting for possible endogeneity and heteroskedasticity between charitable giving and R&D innovation.

**Keywords:** charitable donation; R&D innovation; corporate life cycle; high-tech enterprises

## 1. Introduction

Innovation is the lifeline of high-tech enterprises [1]. The U.S.–China technology battle shows that, without mastering core technologies, seemingly powerful companies will instantly fall into a life-or-death situation. Mastery of core technologies requires firms to carry out continuous R&D activities. Improving R&D capabilities has become a key source and driver of competitive advantage in all industries [2,3]. However, as integration and synergy among different elements are dynamic, focusing only on professional human capital and technical capital may not confer distinct technological advantages to high-tech enterprises. A strong brand reputation and close cooperation among multiple stakeholders such as employees and technology partners are also critical. Especially under the dual impact of the current world turmoil and change across industries and the COVID-19 pandemic, high-tech enterprises are facing external technology disruption and continuous supply chain crisis. Therefore, building a solid stakeholder network is necessary to resist the impact of external crisis and enhance organizational resilience. Without close collaboration among suppliers, customers, and other stakeholders to weather these difficulties, no firms can survive under the impact of the external crisis.

Research has shown that charitable giving not only has its social responsibility dimension [4], but also is an effective way for firms to exchange resources with stakeholders and gain social capital [4–6], such as a good brand image [7], attraction and retention of high-quality employees [8], and accumulation of relationship assets [9,10]. By constructing

an ecosystem of social responsibility, firms form a "symbiotic relationship" with their stakeholders [11], thereby enhancing R&D innovation and organizational resilience. Charitable giving is as important as R&D innovation for firms to gain competitive differentiation and achieve sustained growth [12,13].

As an important strategy for gaining competitive advantage, the interactive behavior of charitable giving and R&D innovation in strategy execution and strategic benefits has triggered extensive discussions among scholars on the relationship between them. However, the relationship between them is still controversial at this stage, both in terms of theoretical explanations and empirical tests. Based on resource-based theory, some scholars believe that the resources available for discretionary use within a firm are limited and that implementing charitable giving and R&D innovation will lead to a "crowding-out" effect as they compete with each other in the firm's resource market [13–16]. Other scholars, based on stakeholder theory, believe that charitable giving helps firms to build a positive image [17], access external scarce resources [18], and build government–enterprise ties to mitigate the negative impact of policy uncertainty on R&D innovation [19], generating a "synergy" effect [20–26]. In addition, some scholars believe that the relationship between them is not directly linear, but a non-linear U-shaped and inverted U-shaped relationship [27,28].

A possible explanation for the inconclusive findings is that the relationship between charitable giving and R&D innovation is not a single static relationship, but can exhibit a dynamic process of development at different stages. This relationship is a more complex relationship than depicted by current studies. While firm resource capacity is an important factor influencing the interaction between charitable giving and R&D innovation [29], existing studies only tend to examine the impact of resource-level differences on the relationship within a single time dimension, ignoring the divergent resource levels of firms at different life cycle stages. Specifically, both charitable giving and R&D innovation require firm resources, but firms face different opportunities and challenges at different life cycle stages [30], and have different strategic objectives. Firms prioritize maximizing their portfolio of investments in conjunction with their current strategic objectives, leading to significant differences in the acquisition, integration, and allocation of resources and the scope of their main business activities. As a result, their motivation for R&D and charitable investment and the "concurrence" relationship between them changes. Therefore, based on the gaps in existing research, this paper introduces the corporate life cycle as a contextual variable to explore the impact of life cycle differences on the relationship between charitable giving and R&D innovation based on resource-based theory and stakeholder theory. Based on this, this paper will seek to answer the following two questions:

Q1: Is there a significant correlation between charitable giving and R&D innovation?

Q2: Do different life cycle stages affect the interaction between charitable giving and R&D innovation?

Therefore, this study selects high-tech enterprises listed on Chinese A-shares from 2015 to 2020 as the sample data for empirical analysis. The findings of the study provide a unique perspective to explain the theoretical controversy and the uncertainty of the empirical results between charitable giving and R&D innovation, and also have important practical implications for high-tech enterprises to maximize the value of firm resources and enhance organizational resilience according to their different stages of development.

The remainder of this paper is organized as follows. The following section reviews background literature, including the development of research assumptions. Section 3 describes the data, method, and statistical techniques used in this study. Section 4 presents the empirical results obtained, including robustness checks. Section 5 addresses further analysis of the synergy relationship between charitable giving and R&D innovation. Section 6 discusses the present study's findings and some possible future research directions. The concluding section summarizes and concludes the study.

## 2. Literature Review and Research Assumptions

### 2.1. Charitable Giving and R&D Innovation

As important strategic choices for firms, the interaction between charitable giving and R&D innovation is signaled through firm resources, and the study of the relationship between them is actually a study of the issue of firm resources [5].

In the overall strategic layout of a firm, the internal R&D strategy is crucial. In today's competitive environment, R&D innovation and professional capital are useful for product differentiation to create inimitable resources and build core competitiveness in high-tech enterprises for sustained growth [1]. However, with the current prominent tail effect of the COVID-19 pandemic and rising global commodity prices, the R&D input and output of firms have shown an increasingly weak correlation. In addition, R&D innovation is facing extremely high technical, market, and financial risks. To survive in this landscape, firms urgently need a good policy environment, easy access to financing, and other social capital to mitigate external institutional risks and uncertainties [31]. In addition, strong social capital is a stabilizer and ballast for firms to conduct R&D innovation. Based on stakeholder theory, scholars from the "promotion" viewpoint believe that charitable giving not only has its own social responsibility dimension, but is also an important way for firms to build social capital [4]. The benefits of this important resource includes a philanthropic image for firms, enhanced brand premiums [32], fewer financing barriers [33], access to scarce government resources, legitimacy recognition, and a government–firm cooperation platform to bring the government–firm linkage effect [34]. Consequently, firms will resolve various resource and policy risks to the R&D process, reduce the cost of R&D innovation, and finally form a strategic synergy between social capital and professional capital [4].

On the contrary, scholars [24] with an "inhibition" view argue, based on resource-based theory, that charitable giving and R&D innovation, as long-term firm strategies, focus on creating long-term strategic benefits for the firm. Therefore, both charitable giving and R&D innovation require sustained resource investment by the firm to trigger the scale effect. For high-tech enterprises, the economic downturn has exacerbated the pressure on internal resources and reduced discretionary resources, inevitably leading to crowding out of internal resources available for implementation of both strategies, thus triggering a competitive dynamic [13]. In addition, the existence of the principal agent problem in firms will trigger the self-interest motivation of managers. They will focus on accumulating individual social capital through charitable giving to achieve personal reputation, career development, and social status, which will eventually lead to excessive donations, thus intensifying the crowding out of R&D innovation resources.

Overall, synergy between charitable giving and R&D innovation may be reflected in both the complementary relationship between social and professional capital and the crowding-out effect due to resource constraints during strategy execution [22]. Based on this analysis, two competing hypotheses are proposed:

**Hypothesis 1a (H1a).** *There is a positive "synergetic" effect between charitable giving and R&D innovation.*

**Hypothesis 1b (H1b).** *There is a negative "crowding-out" effect between charitable giving and R&D innovation.*

### 2.2. The Corporate Life Cycle, Charitable Giving, and R&D Innovation

2.2.1. The Division of the Corporate Life Cycle

Different classifications of the life cycle of firms are provided in the literature. The most common ones include the univariate method [35], the aggregative indicator method [36], and the cash flow method [37]. However, the linear progression assumption of the univariate method for a firm's life cycle does not conform to the dynamic law of the firm's life cycle development and only provides partial information about the firm life cycle. In contrast, the selection of the underlying variables of the aggregative indicator method fails

to capture all attributes of the firm life cycle [38]. Dickinson [37] pointed out that, with the cash flow method, it is easier to identify differences in profitability performance at different life cycles than when using the univariate method or the aggregative indicator method. In addition, the cash flow method has better consistency with the economic theory of the firm. Thus, this paper uses the Dickinson cash flow method to identify the corporate life cycle.

Dickinson [37] decomposed cash flows into operating cash flows, investment cash flows, and financing cash flows based on different forms of firm cash flows to reflect differences in profitability, growth capacity, and risk capacity, respectively. He also divided the life cycle into five stages: start-up, growth, maturity, shock, and decline, based on different combinations of these three cash flows. From these life cycle stages, the actual development stage of Chinese A-share listed companies is beyond the start-up period. Therefore, drawing on the research of Xie and Wang [39] and Chen et al. [40], this paper merges the start-up and growth periods into the growth stage while the mature period and the sample in the shock period with similar characteristics to the maturity firms are classified as the mature stage; in addition, the decline period and the sample in the shock period with similar characteristics to the declining firms are classified as the decline period. The specific classification and the cash flow characteristics of firms at different life cycle stages are shown in Table 1 below.

**Table 1.** Types of cash flow combinations in different life cycles.

| Category | Growth | | Mature | | | | Decline | |
|---|---|---|---|---|---|---|---|---|
| Operating cash flow symbol | − | + | + | − | + | + | − | − |
| Investment cash flow symbol | − | − | − | − | + | + | + | + |
| Financing cash flow symbol | + | + | − | − | + | − | + | − |

Notes: "+" represents a positive net cash flow and "−" represents a negative net cash flow.

### 2.2.2. Growth Firms, Charitable Giving, and R&D Innovation

The primary goal of the growth stage of firms is to achieve expansion and growth [41]. During this period, the demand for capital expands dramatically, but the production process and scale of the product cannot guarantee continuous and stable cash flow growth for the firm. This limitation, coupled with the low self-accumulation ability of high-tech enterprises in the growth stage and the late start of the venture capital industry in China, exacerbates the shortage of funds, causing a dilemma for the growing firms. With multiple market investment opportunities [42], the primary issue for management is how to allocate the limited financial resources among different strategic projects to help the firm achieve the largest growth in consumer groups and market share in the short-term. Compared with charitable giving, investment in R&D helps firms achieve short-term strategic goals by directly improving and creating products and their production processes. Therefore, growth-stage firms are more likely to use their cash holdings to expand product research than make charitable giving. Campbell [29] suggests that, from an economic perspective, when firms are in the process of discovering "new things", management and external stakeholders will focus more on areas that maximize capital utilization in the short-term and bring substantial profitability to the firm. For example, firms will devote more resources to innovative activities that will lead to profitable growth to capture market opportunities, but choose to be "blinded for a short time" to potential strategic benefits such as charitable giving [43]. Choosing to use their limited resources for charitable giving to achieve a positive brand image will undoubtedly exacerbate internal resource constraints for growing firms with limited resources. Coupled with the relatively weak combination of firm capital at this stage, it is difficult for charitable giving to trigger returns of scale in the short term before a certain accumulation is formed to make a large contribution to R&D innovation. Some scholars even argue that the more resources invested in R&D innovation, the less firms will donate to help the public and solve social problems [44].

In addition, Carroll [45] proposes a hierarchy of corporate social responsibility, including economic, legal, ethical, and philanthropic dimensions. In this model, charitable giving based on altruistic motives is only truly implemented after a firm has achieved the economic, legal, and ethical goals of its own operations. As a result, there are significant differences in the social responsibilities that firms at different stages of economic development should fulfill [39]. During the growth period, the market for the firm's products has not yet matured and the firm still experiences certain limitations in getting rid of financial constraints. At this time, consumers and other external stakeholders focus more on product sales and quality, and less on the social responsibility activities of the firm, which primarily aim to establish good brand preferences and expand product sales in the minds of consumers. Excessive charitable giving at this time is undoubtedly a "thankless" behavior, which will not only intensify the competition for resources between firm strategies and reduce the utilization of resources, but also easily trigger opposition from stakeholders and bring policy risks for R&D innovation [28].

In conclusion, as growth-stage firms have not yet reached their own financial goals of operation, their R&D budget is limited and based on the need for new product development and market expansion [38]. Even if some firms make charitable giving to pursue advertising effect and brand image, for the growth-stage firms with limited resources, such actions are likely to trigger opposition from stakeholders, affecting R&D innovation. Based on this analysis, the following hypothesis is proposed:

**Hypothesis 2 (H2).** *Charitable giving by growth-stage firms is not effective in promoting higher levels of R&D innovation.*

### 2.2.3. Mature Firms, Charitable Giving, and R&D Innovation

Mature firms are known as "cash cows"; they not only have stable cash flow, but also a fixed consumer group in the product market and a certain level of popularity in the social market. At this stage, firms begin to pursue different goals that distinguish them from growth-stage firms [46]. First, according to Maslow's Hierarchy of Needs theory, mature firms have basically satisfied their "security needs"; specifically, their financial security needs. They turn to pursuing higher-level "social needs", including building a more diversified network of stakeholders, such as government, community, and hidden consumers [30]. As an important means of exchanging resources with external stakeholders [47], charitable giving has become a key strategy for overcoming the barriers to higher-level needs. At the maturity stage, the firm's product sales and profits have grown considerably, its capital is relatively abundant, and its organizational decisions are less constrained by resources. Therefore, the implementation of charitable giving may not seriously crowd out resources for R&D innovation. Secondly, high-tech firms choose to gain competitive advantage by implementing differentiation strategies, such as charitable giving and R&D innovation, to meet the growing heterogeneity of consumer demand and the profitability goals of other external stakeholders [48]. The resource-based view suggests that the use of charitable giving as a differentiation strategy depends on the life cycle of the firm. For firms at the growth stage, the use of differentiation strategies to gain competitive advantage is limited [20], whereas for firms in the maturity stage, charitable giving is pursued to gain social benefits and a favorable brand image [23]. Charitable giving is pursued to achieve an advertising effect to promote a firm's new products to the market, while driving the firm to develop more ecological and environmentally friendly products and services that meet the needs of social development [13]. Likewise, R&D innovation drives firms to continue their charitable giving, making this socially responsible behavior more technological and efficient [49]. Ultimately, charitable giving and R&D innovation influence and promote each other, which in turn triggers the strategic linkage of social and professional capital to achieve new growth and leapfrogging of firms [1].

In addition, mature firms have passed the rapid growth stage and begin to enter the platform stabilization stage, when they begin to focus on other goals besides financial growth, such as social impact and reputation [38]. At the same time, the demands by consumers and other stakeholders for corporate responsibility as social citizens begin to grow, and donation behavior at this time flows more. At this stage, charitable giving spreads the effect among social groups and fosters a good reputation for the firm. It also enhances brand value, stimulates the pride of R&D employees, and can even gain the support of external R&D partners and government R&D funding subsidies to R&D innovation, ultimately yielding the demonstration effect of "charitable for innovation" [26].

In conclusion, the financial resources of mature firms are relatively abundant, and the occurrence of donation behavior does not seriously crowd out resources for R&D innovation, but rather helps firms build a good reputation, enhance brand value, obtain scarce resources, and inject growth vitality into R&D innovation [50]. Based on this analysis, the following hypothesis is proposed:

**Hypothesis 3 (H3).** *Charitable giving by mature firms can effectively promote the level of R&D innovation.*

2.2.4. Declining Firms, Charity Giving, and R&D Innovation

Declining firms have a significant advantage over growth firms, including cash holdings accumulated from past operations. However, the firm's existing products gradually begin to lose market competitiveness, resulting in a year-by-year decline in market share and profitability [49]. To avoid market share loss or closure of operations due to weakening product competitiveness and poor management, firms must continuously expand investment in R&D to develop new products and explore new market opportunities [31]. However, the development of new products requires complex external resources, including corporate reputation, long-term rapport with suppliers and customers, and technical knowledge [50]. For declining firms, charitable giving as a benefit exchange tool between the firm and external stakeholders can provide strategic support for R&D activities [18,51], including establishing a charitable image to attract the attention of media and consumers [11]. It can also send positive signals to the outside world that the business is doing well to alleviate stakeholders' sense of crisis, continue to attract investment [5], and bring external information and scarce resource supply to support the firm's subsequent product innovation and explore new market opportunities [4].

Overall, the financial situation of declining firms is significantly better than that of growing firms. As a result, declining firms still have the capacity to support donation without crowding out the resources needed for R&D innovation [49]. Based on this analysis, the following hypothesis is proposed:

**Hypothesis 4 (H4).** *Charitable giving by declining firms can effectively promote the level of R&D innovation.*

The research model is shown in Figure 1, which maps the hypothesized associations among corporate giving, R&D innovation, and the influence of different corporate life cycle stages on the relationship between them.

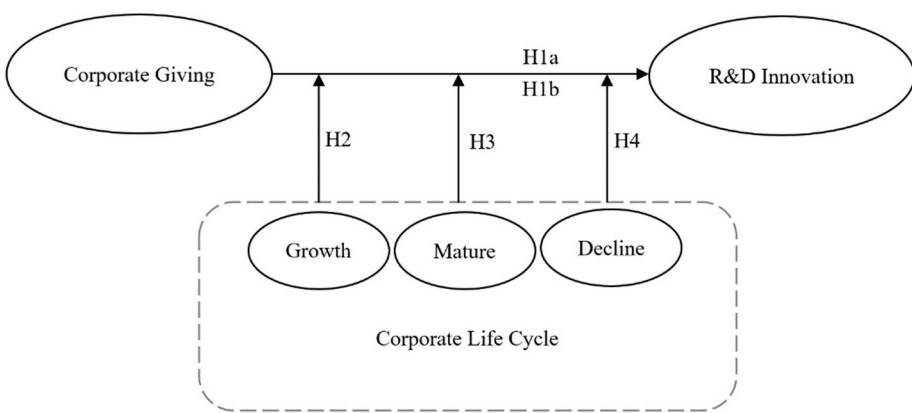

**Figure 1.** Research model.

## 3. Research Methodology

### 3.1. Data

This paper selects high-tech firms listed in Chinese A-shares from 2015 to 2020 as the research object. A firm is classified as a high-tech enterprise based on the statistical criteria of the qualification recognition information of listed firms in China disclosed in the database of the China Stock Market & Accounting Research Database (CSMAR). The CSMAR is an accurate research-oriented database in the economic and financial field developed by Shenzhen Xishima Data Technology Co., Ltd. It is designed for academic research needs, drawing lessons from CRSP, COMPUSTAT, TAQ, THOMSON, and other authoritative database professional standards and combining the actual situation of China. It is also a data source for more than 500 universities and financial institutions in China for empirical analysis on securities, economics, and finance. Considering that firms achieve the recognition of high-tech enterprises in China after three years of operation, it is necessary to recognize the identity of high-tech enterprises after the expiration of the period. Therefore, data within three years after their recognition were selected as the sample period [52].

The sample data in this paper were obtained from the CSMAR, and the econometric analysis was performed using Stata16. A total of 14,684 initial sample values were obtained from 3228 firms. The initial sample was screened as follows: (1) ST and *ST firms were excluded; (2) samples with missing values for key data such as R&D innovation and charitable giving were excluded; (3) observations where the cash flow of listed firms for the current year was zero and other indicators were not available were excluded. Finally, 9863 valid samples were obtained. Considering the possible influence of extreme values on the study results, all continuous variables were winsorized at 1% to 99% quartiles [5,53].

### 3.2. Variable Definition and Measurement

#### 3.2.1. R&D Innovation

R&D innovation (R&D) is the dependent variable used in this study. Referring to the research of Makri et al. [54], the R&D innovation was measured by the ratio of R&D investment and the sales revenue of the firm in the current year. At the same time, in order to reduce the impact of earnings management and new product pricing strategies on the indicators, "lg(R&D investment)" was used as an alternative variable to measure R&D innovation.

#### 3.2.2. Charitable Giving

Charitable giving (Don) is the independent variable used in this study. The relevant data were obtained from the CSMAR with the word "donation" in the specific items of non-operating expenses disclosed. Data with "non-public welfare" and "donation and penalty expenses", which are obviously unrelated to charitable giving, were excluded. For the samples with missing donation data, the amount of firm giving disclosed in the

CSR database was used. Based on existing reports [55], the ratio of charitable giving to sales revenue was chosen as a measure of the level of charitable giving, and "lg(charitable giving)" was chosen as its alternative variable. Since this sample set had no firm with a donation of 0, all data were processed logarithmically.

### 3.2.3. Corporate Social Responsibility (CSR)

In this study, firms with growth characteristics for 5 years or more within 6 years were grouped as growth-stage firms (Growth) and firms with mature characteristics for 5 years or more within 6 years as mature-stage firms (Mature). Considering that the decline characteristics of high-tech firms may not be significant, this study classified the firms with decline characteristics for 4 years or more within 6 years as decline-stage firms (Decline), and the remaining firms with unstable life cycle characteristics as the control group (Control) for comparative study [37]. The sample numbers of different corporate life cycles obtained in this paper are shown in Table 2. As expected, the number of decline-stage firms was the lowest (7.8%). However, the number of growth-stage firms (59.4%) was much higher than the number of mature-stage firms (32.8%), indicating that Chinese high-tech firms are in a growing development stage.

**Table 2.** The number of samples in different life cycles.

|                   | Control | Growth | Mature | Decline |
|-------------------|---------|--------|--------|---------|
| Number of firms   | 2721    | 114    | 63     | 15      |
| Number of samples | 8763    | 653    | 365    | 82      |

### 3.2.4. Control Variables

In order to reduce the impact of other variables on R&D innovation, this study introduces government subsidy intensity (Sub), firm size (Size), firm age (Age), degree of financial leverage (DFL), return on total assets (ROA), ownership concentration (OC), financial slack (FS), CEO duality (Dual), Tobin Q (Tobin's Q), and dummy variables for industry (In), year (Ye), and province (Pr) as control variables [5,13,56]. Of them, the amount of government subsidies is derived from the sum of the amount of government subsidies in other income and non-operating income. The specific variable definition method is shown in Table 3.

### 3.3. Statistical Techniques

This study used a quantitative research method. Winsorized continuous variables cause the model data to exhibit the typical characteristics of censored data, making it difficult to achieve consistent estimation using OLS regression. The Tobit model using MLE estimation can help to solve such censored data problems (after testing the correlation model with the xttest3 command, the original hypothesis of homoscedasticity was strongly rejected; additionally the original hypothesis of "no intra-group autocorrelation" was strongly rejected after testing with the xtserial command) [57]. This study used clustering of robust standard errors to mitigate heteroskedasticity and intra-group autocorrelation problems. Meanwhile, the estimation results of the Fixed Effects models and Random Effects models are reported. The specific econometric models are set as follows (In the process of model setting, in order to reduce the impact of the model's "setting error" on the estimation results, this study fully considers the problem of omitted variable bias and follows the modeling strategy of "general to specific". Firstly, all possible independent variables were collected and insignificant independent variables were gradually eliminated; secondly, the panel data analysis and the Instrumental Variables Method used in this paper in the regression analysis both helped to avoid possible omitted variable bias in the model and improve the robustness of the estimation results):

$$Y_{i,t} = \beta_0 + \beta_1 X_{i,t} + \sum \beta_k \text{Controls}_{i,t} + \gamma_{In} + \mu_{Ye} + \varphi_{Pr} + \varepsilon_{i,t} \tag{1}$$

where i represents the firm ID and t represents the year in which the firm is located. The dependent variables $Y_{i,t}$ on the left side of the equation are R&D1 and R&D2, representing the intensity of firm R&D innovation. $\beta_0$ on the right side of the equation is a constant term. The independent variables $X_{i,t}$ are Don1 and Don2, representing the intensity of firm charitable giving, whose signs and coefficient magnitudes can be used to identify the effect of charitable giving on R&D innovation. Controls is a set of control variables in this paper, while controlling for the effects of non-time-varying industry ($\gamma_{In}$), year ($\mu_{Ye}$), and regional economy ($\varphi_{Pr}$) on the model. $\varepsilon$ is the random disturbance term.

**Table 3.** Table of variable definitions.

| Types of Variables | Variable Name | Variable Code | Variable Definition |
|---|---|---|---|
| Dependent variable | R&D innovation | R&D1 | R&D investment/Sales revenue |
| | | R&D2 | Lg(R&D investment) |
| Independent variable | Charitable giving | Don1 | Charitable giving/Sales revenue |
| | | Don2 | Lg(Charitable giving) |
| Grouping variable | Corporate life cycle | CLC | It is determined by the Dickinson cash flow group method, as shown in Table 1 |
| Control variable | Government subsidy | Sub | Government subsidy/Sales revenue |
| | Firm scale | Size | Ln(Total assets) |
| | Firm age | Age | Year of the sample-Year of the firm's IPO |
| | Financial leverage | DFL | Total liability/Total assets |
| | Return on total assets | ROA | Net profit/Total assets |
| | Ownership concentration | OC | Share ratio of the largest shareholder/The total number of shares |
| | Financial slack | FS | Operating cash flow/Sales revenue |
| | CEO duality | Dual | If the chairman and the general manager are held by one person, the value is 1; otherwise, the value is 0 |
| | Tobin Q | Tobin's Q | Total value/Total assets |
| | Industry | In | Industry dummy variable: the data for this industry is 1; otherwise, the value is 0 |
| | Year | Ye | Year dummy variable: the data for this industry is 1; otherwise, the value is 0 |
| | Province | Pr | Province dummy variable: the data for this industry is 1; otherwise, the value is 0 |

## 4. Empirical Results

### 4.1. Descriptive Statistics

Table 4 shows the descriptive statistics analysis of the variables. From the table, the mean values of R&D innovation (R&D) are 0.055 and 18.307, with standard deviations of 0.046 and 1.329, respectively, indicating significant differences in R&D innovation among different firms; the mean values of charitable giving (Don) are 0.001 and 12.808, and the median values are 0.000 and 12.869, respectively, in line with the normal distribution characteristics; the mean value of government financial subsidies (Sub) is 0.016, while the maximum value reaches 0.1, indicating that a few firms receive strong financial support from the government; the mean value of firm size (Size) is 22.273, and the standard deviation is 1.236, indicating that the size difference between different firms is reasonably distributed and mostly listed after 2000, which is in line with the growth status of high-tech enterprises; and the distribution of the rest of the variables is within a reasonable range.

**Table 4.** Descriptive statistics analysis.

| Variables | Number | Mean | Std. Dev | Min | Median | Max |
|---|---|---|---|---|---|---|
| R&D1 | 9863 | 0.055 | 0.046 | 0.000 | 0.044 | 0.262 |
| R&D2 | 9863 | 18.307 | 1.329 | 0.000 | 18.204 | 24.104 |
| Don1 | 9863 | 0.001 | 0.001 | 0.000 | 0.000 | 0.007 |
| Don2 | 9863 | 12.808 | 2.016 | 0.732 | 12.869 | 23.719 |
| Sub | 9863 | 0.016 | 0.018 | −0.005 | 0.010 | 0.100 |
| Size | 9863 | 22.273 | 1.236 | 18.615 | 22.098 | 28.416 |
| Age | 9863 | 18.427 | 5.551 | 3.000 | 18.000 | 62.000 |
| DFL | 9863 | 0.398 | 0.187 | 0.014 | 0.390 | 0.842 |
| ROA | 9863 | 0.042 | 0.067 | −0.308 | 0.043 | 0.208 |
| OC | 9863 | 0.322 | 0.140 | 0.029 | 0.301 | 0.724 |
| FS | 9863 | 0.102 | 0.136 | −0.390 | 0.099 | 0.481 |
| Dual | 9863 | 0.337 | 0.473 | 0.000 | 0.000 | 1.000 |
| Tobin's Q | 9863 | 2.196 | 1.954 | 0.048 | 1.618 | 11.100 |

*4.2. Correlation Analysis*

The correlation coefficient matrix in Table 5 shows that the correlation coefficient between charitable giving (Don) and R&D innovation (R&D) is 0.146 and highly significant, which tentatively verifies hypothesis H1a. Meanwhile, the correlation coefficient between the independent variables is less than 0.3, and the maximum value of the variance inflation factor (VIF) is 1.8 and the mean value is 1.28, which is much less than 10, basically excluding the problem of multicollinearity.

**Table 5.** Correlation matrix for major variables.

| Variables | R&D1 | Don1 | Sub | Size | Age | DFL | ROA |
|---|---|---|---|---|---|---|---|
| R&D1 | 1 | | | | | | |
| Don1 | 0.146 *** | 1 | | | | | |
| Sub | 0.510 *** | 0.114 *** | 1 | | | | |
| Size | −0.243 *** | −0.130 *** | −0.186 *** | 1 | | | |
| Age | −0.128 *** | −0.009 | −0.092 *** | 0.128 *** | 1 | | |
| DFL | −0.275 *** | −0.200 *** | −0.185 *** | 0.535 *** | 0.108 *** | 1 | |
| ROA | −0.001 | 0.081 *** | 0.013 | −0.065 *** | −0.051 *** | −0.357 *** | 1 |

Note: *** $p < 0.01$.

*4.3. Hypothesis Testing*

4.3.1. Analysis of the Relationship between Charitable Giving and R&D Innovation

Table 6 shows the regression results of the interaction between charitable giving (Don) and R&D innovation (R&D). Where columns (1) and (2) correspond to the estimation results of the Fixed Effects models, columns (3) and (4) correspond to the estimation results of the Random Effects models, and columns (5) and (6) correspond to the estimation results of the Tobit models. In addition, the robustness of the findings is further enhanced by replacing key variables in this paper, where R&D1 and Don1 are chosen as key variables in columns (1), (3), and (5), and R&D2 and Don2 are chosen as key variables in columns (2), (4), and (6). As shown in Table 6, the Tobit model corresponds to the two sets of estimated coefficients of 2.185 ($p < 0.01$) and 0.027 ($p < 0.01$), respectively, with slight differences between the estimated coefficients affected by variable measurement methods; however, both are highly significant at the level of 1%. The estimation results of the remaining two econometric models are consistent, indicating a significant positive relationship between charitable giving and R&D innovation. The donation behavior results in scarce policy resources, good reputation, and a convenient financing environment for high-tech enterprises, which enhances organizational resilience and enables firms to have enough buffer space to cope with the strong impact of technology competition and the COVID-19 pandemic shock on R&D innovation. In conclusion, hypothesis H1a is verified.

**Table 6.** Charitable giving and R&D innovation.

| Variables | (1) | (2) | (3) | (4) | (5) | (6) |
|---|---|---|---|---|---|---|
| | **Fixed Effect Model** | | **Random Effect Model** | | **Tobit Model** | |
| | **R&D1** | **R&D2** | **R&D1** | **R&D2** | **R&D1** | **R&D2** |
| Sub | 0.402 *** | −1.105 * | 0.600 *** | 0.679 | 0.959 *** | 1.958 *** |
| | (9.266) | (−1.904) | (14.507) | (1.255) | (18.530) | (2.715) |
| Size | 0.000 | 0.716 *** | −0.000 | 0.787 *** | 0.002 *** | 0.902 *** |
| | (0.268) | (20.178) | (−0.330) | (38.790) | (3.212) | (56.399) |
| Age | 0.002 *** | 0.084 *** | 0.000 *** | 0.030 *** | −0.000 *** | −0.001 |
| | (6.918) | (13.907) | (3.686) | (10.688) | (−3.399) | (−0.503) |
| DFL | −0.022 *** | −0.182 | −0.031 *** | −0.219** | −0.030 *** | 0.044 |
| | (−4.067) | (−1.590) | (−7.346) | (−2.346) | (−7.329) | (0.484) |
| ROA | −0.044 *** | −0.182 | −0.052 *** | −0.226 ** | −0.055 *** | 0.955 *** |
| | (−6.620) | (−1.536) | (−8.012) | (−2.022) | (−5.891) | (5.186) |
| OC | 0.010 | 0.434 | −0.029 *** | −0.248 | −0.022 *** | 0.043 |
| | (1.282) | (1.382) | (−5.960) | (−1.454) | (−5.456) | (0.458) |
| FS | −0.029 *** | −0.129 ** | −0.023 *** | −0.065 | −0.021 *** | −0.212 ** |
| | (−10.082) | (−2.413) | (−8.711) | (−1.292) | (−4.384) | (−2.328) |
| Dual | 0.000 | 0.002 | 0.002 ** | 0.023 | 0.002 | −0.023 |
| | (0.461) | (0.069) | (2.371) | (0.855) | (1.630) | (−0.933) |
| Tobin's Q | −0.000 | 0.023 *** | 0.001** | 0.012 *** | 0.005 *** | 0.057 *** |
| | (−0.394) | (4.722) | (2.366) | (3.082) | (9.000) | (7.813) |
| Don1 | 1.312 *** | | 1.961 *** | | 2.185 *** | |
| | (3.880) | | (5.739) | | (4.190) | |
| Don2 | | 0.008 * | | 0.019 *** | | 0.027 *** |
| | | (1.755) | | (4.149) | | (4.407) |
| In/Ye/Pr | Yes | Yes | | | Yes | Yes |
| Constant | 0.017 | 0.596 | 0.068 *** | 0.085 | 0.007 | −2.614 *** |
| | (0.502) | (0.778) | (4.121) | (0.197) | (0.262) | (−5.495) |
| F | 22.174 | 178.702 | | | | |
| R2_within | 0.143 | 0.428 | 0.110 | 0.401 | | |
| Pseudo R2 | | | | | −0.220 | 0.357 |
| N | 9863 | 9863 | 9863 | 9863 | 9863 | 9863 |

Note: Robust *t*-statistics in brackets; * $p < 0.1$, ** $p < 0.5$, *** $p < 0.01$.

### 4.3.2. Analysis of the Impact of CSR on the Relationship between Charitable Giving and R&D Innovation

Table 7 shows the results of the impact of different life cycle stages on the relationship between charitable giving (Don) and R&D innovation (R&D). As shown in Table 7, the synergistic utility between charitable giving and R&D innovation varies widely across life cycle stages. First, as a reference, for the control group firms, the regression coefficient of Donation1 is 1.854 ($p < 0.01$) and there is a significant positive correlation between them. The regression results after distinguishing different life cycles show that the synergy between charitable giving and R&D innovation is most significant for mature firms ($\beta = 12.709$, $p < 0.01$), followed by declining firms ($\beta = 3.358$, $p < 0.05$), while for growing firms, the synergy was not significantly supported ($\beta = 2.31$). Maturity, as the glorious period in a firm's life cycle, is characterized by sufficient cash flow and peak capital and technology levels. High-tech enterprises are strong and resilient enough to cope with external environmental shocks and can guarantee the continuous operation of R&D processes through charitable giving, which complement each other. On the contrary, the resource capacity and development level of growth-stage firms cannot support the differentiated competitive approach of charitable giving to secure key scarce resources for R&D innovation. In conclusion, hypotheses H2, H3, and H4 are all verified in this paper, and graphs are drawn to visualize the findings, as shown in Figure 2 below.

**Table 7.** The impact of CSR on the relationship between charitable giving and R&D innovation.

| Variables | Tobit Model | | | |
|---|---|---|---|---|
| | R&D1 | | | |
| | **(1)** | **(2)** | **(3)** | **(4)** |
| | Control | Growth | Mature | Decline |
| Sub | 0.954 *** | 0.912 *** | 0.446 * | 0.063 |
| | (19.332) | (4.873) | (1.952) | (1.086) |
| Size | 0.002 *** | −0.005 * | 0.007 ** | 0.016 *** |
| | (3.106) | (−1.830) | (2.430) | (3.302) |
| Age | −0.000 *** | −0.002 * | −0.002 *** | −0.002 *** |
| | (−2.827) | (−1.788) | (−4.756) | (−3.504) |
| DFL | −0.032 *** | −0.011 | −0.055 *** | 0.006 |
| | (−7.298) | (−0.593) | (−2.929) | (0.353) |
| ROA | −0.055 *** | −0.046 | −0.111 *** | −0.024 * |
| | (−5.611) | (−1.129) | (−2.762) | (−1.975) |
| OC | −0.022 *** | −0.031 | 0.014 | −0.073 |
| | (−5.419) | (−1.202) | (0.677) | (−1.561) |
| FS | −0.022 *** | −0.024 | −0.039 *** | −0.007 |
| | (−4.335) | (−1.323) | (−3.212) | (−1.429) |
| Dual | 0.002 * | −0.002 | −0.002 | −0.004 |
| | (1.744) | (−0.298) | (−0.462) | (−1.003) |
| Tobin's Q | 0.005 *** | 0.003 * | 0.001 | 0.003 * |
| | (8.989) | (1.900) | (0.840) | (1.730) |
| Don1 | 1.854 *** | 2.310 | 12.709 *** | 3.358 ** |
| | (3.502) | (1.330) | (3.922) | (2.432) |
| In/Ye/Pr | Yes | Yes | Yes | Yes |
| Constant | 0.007 | 0.149 ** | −0.097 | −0.289 *** |
| | (0.263) | (2.401) | (−1.547) | (−2.967) |
| Pseudo R2 | −0.222 | −0.395 | −0.380 | −0.345 |
| N | 8763 | 653 | 365 | 82 |

Note: Robust *t*-statistics in brackets; * $p < 0.1$, ** $p < 0.5$, *** $p < 0.01$.

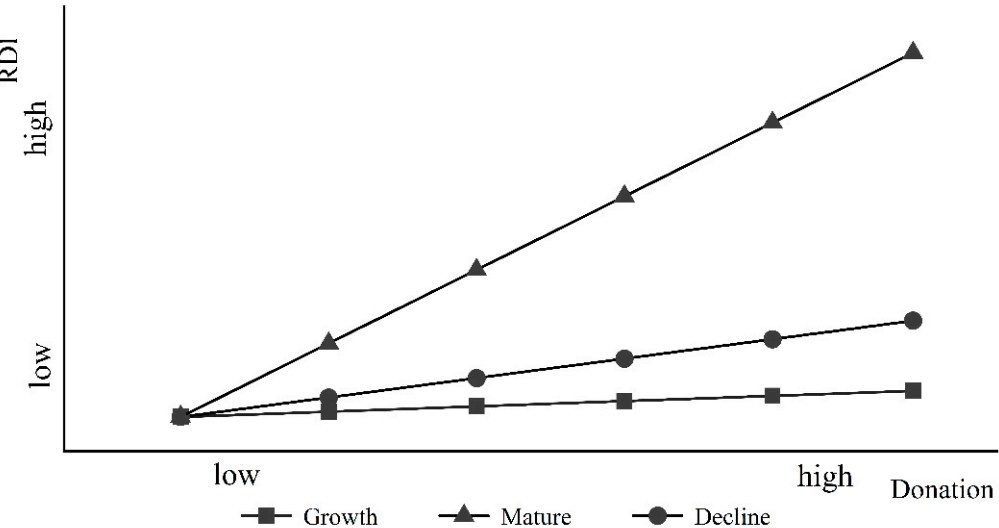

**Figure 2.** Utility differences between charitable giving and R&D innovation at different life cycle stages.

4.3.3. Endogeneity Test (2SLS)

In terms of the logic of interaction between charitable giving and R&D innovation, there can be endogeneity between them that arises from mutual causation [58]. On the one hand, the social capital brought by charitable giving to firms can largely help them obtain government resource support for R&D innovation. Additionally, the accumulation of professional capital resulting from R&D innovation can bring new profit growth for firms and increase the frequency of charitable giving [58]. The existence of endogenous problems can make the estimation results biased. To further confirm whether the independent variables are endogenous, a Hausman test was conducted, and the *p*-value of model 1 in Table 8 was 0.043, rejecting the original hypothesis that "all independent variables are exogenous" at the 5% level, which means that charitable giving is considered endogenous. To solve the problem, the Two-Stage Least Square method (2SLS) is used for regression analysis (the basic idea is to regress the instrumental variables with endogenous independent variables in the first-stage regression to obtain the fitted value $\hat{p}$, where the fitted value $\hat{p}$ represents the part of the endogenous variables that are not correlated with the disturbance term, and then regress the fitted value of the first-stage regression with the independent variables in the second stage; then, the problem of endogenous independent variables can be solved), and the regression model is shown below.

$$\Phi(Z_{i,t}) = \text{Donation1}_{i,t} = \alpha_0 + \alpha_1 \text{IV1}_{i,t} + \alpha_2 \text{IV2}_{i,t} + \gamma_{IN} + \mu_{YE} + \varphi_{PR} + \delta_{i,t} \tag{2}$$

$$\text{RDI1}_{i,t} = \beta_0 + \beta_1 Z_{i,t} + \sum \beta_k \text{Controls}_{i,t} + \gamma_{IN} + \mu_{YE} + \varphi_{PR} + \varepsilon_{i,t} \tag{3}$$

For the selection of instrumental variables, firstly, based on the research of Guo [59] and the industry-level macro data, the mean value of industry charitable giving (IV1), was selected as one of the instrumental variables. Mean values of charitable giving at the industry level are correlated with corporate charitable giving, but had little effect on firm R&D innovation, in line with the principles of relevance and exclusivity in the selection of instrumental variables. Secondly, charitable giving with a one-period lag (IV2) was selected as one of the instrumental variables. Obviously, charitable giving is related to its lagged term, but from a current period perspective, current R&D innovation cannot affect past charitable giving (Before conducting 2SLS estimation, the following tests were conducted on the two selected instrumental variables to ensure the validity of the instrumental variable selection. i. Weak instrumental variable test: taking model 1 in Table 8 as an example, the F-value of the first-stage regression is 25.36 (greater than 10), which can basically exclude the problem of weak instruments; secondly, the results of the redundancy test for both instrumental variables strongly rejected the null hypothesis that "IV1 and IV2 are redundant instruments". ii. Overidentification test: all models in Table 8 were tested for override and the null hypothesis of "all instrumental variables are exogenous" was accepted.). The second-stage regression results of 2SLS are shown in Table 8 below (the 2SLS first-stage regression results are not reported in the paper due to space limitations and are available upon request from the authors). The regression results in column (1) show that there is an overall positive relationship between charitable giving and R&D innovation ($\beta = 1.802$, $p < 0.1$). The regression results after distinguishing the different life cycles show that the synergy between charitable giving and R&D innovation is most significant for mature firms ($\beta = 29.723$, $p < 0.01$), followed by declining firms ($\beta = 6.421$, $p < 0.1$), and not significant for growing firms ($\beta = 2.874$). In conclusion, the previous empirical results are generally unaffected by the endogeneity problem.

**Table 8.** The second-stage regression results of 2SLS.

| Variables | R&D1 | | | |
|---|---|---|---|---|
| | **(1)** | **(2)** | **(3)** | **(4)** |
| | **Control** | **Growth** | **Mature** | **Decline** |
| Don1 | 1.802 * | 2.874 | 29.723 *** | 6.421 * |
| | (1.667) | (1.020) | (5.762) | (1.685) |
| Control | Yes | Yes | Yes | Yes |
| In/Ye/Pr | Yes | Yes | Yes | Yes |
| Constant | 0.346 *** | 0.046 | −0.094 | −0.699 *** |
| | (4.819) | (0.682) | (−1.006) | (−2.784) |
| Adj_R2 | 0.375 | 0.573 | 0.380 | 0.728 |
| N | 5612 | 530 | 297 | 63 |

Note: Robust *t*-statistics in brackets; * $p < 0.1$, *** $p < 0.01$.

### 4.3.4. Robustness Test

In the previous regression analysis, the choice of different methods to measure the key variables enhances the robustness of the paper's findings, and the following two approaches are used to conduct robustness tests.

(1) Heteroskedasticity test (Bootstrap): Based on the characteristics of the short panel data, heteroskedasticity is more likely to arise. The heteroskedasticity problem can also lead to biased estimation results. In this paper, a series of tests are conducted to demonstrate the existence of the heteroskedasticity problem (a White test and BP test were performed on all relevant models, and the results both highly rejected the null hypothesis of homoskedasticity at the 1% level, which means that the data were considered to have heteroskedasticity). Although the clustering robust standard errors chosen in the previous regression model can alleviate the heteroskedasticity problem to a certain extent, a more robust Bootstrap self-sampling was chosen to conduct the heteroskedasticity test in this paper. The specific estimation results are shown in Table 9. As seen in column (4) of Table 9, the synergistic utility of charitable giving and R&D innovation is overall positive but not significantly supported for declining firms (β = 3.279). In addition, the remaining findings are consistent with those obtained in previous regression analysis and are not affected by the heteroskedasticity problem.

**Table 9.** Bootstrap model (Bootstrap 1500 times).

| Variables | R&D1 | | | |
|---|---|---|---|---|
| | **(1)** | **(2)** | **(3)** | **(4)** |
| | **Control** | **Growth** | **Mature** | **Decline** |
| Don1 | 1.853 *** | 2.310 | 12.631 *** | 3.279 |
| | (4.422) | (1.622) | (3.781) | (1.088) |
| Control | Yes | Yes | Yes | Yes |
| In/Ye/Pr | Yes | Yes | Yes | Yes |
| Constant | 0.007 | 0.149 *** | −0.097 ** | −0.293 * |
| | (0.462) | (3.781) | (−2.210) | (−1.852) |
| Adj_R2 | 0.514 | 0.672 | 0.746 | 0.750 |
| N | 8763 | 653 | 365 | 82 |

Note: Robust *t*-statistics in brackets; * $p < 0.1$, ** $p < 0.5$, *** $p < 0.01$.

(2) Firms that exhibit growth-stage characteristics for 4 or more years within 6 years are classified as growth-stage firms, whereas those that exhibit mature-stage characteristics for 4 or more years within 6 years are classified as mature-stage firms, and those that exhibit decline-stage characteristics for 3 or more years within 6 years are classified as decline-stage firms. The regression results suggest that the present findings are not affected by the way

the firm is measured at the stage (results are not reported in the text due to space limitations and are available from the authors upon request).

## 5. Further Analysis

To further clarify whether there is heterogeneity in the synergy between charitable giving and R&D innovation at different life-cycle stages, this paper analyzed both the form of ownership and the attributes of senior executives (including their R&D background and their political background) (the data about executive attributes were obtained from the CVs of senior executives in the CSMAR database or character traits study database of listed firms, and dummy variables for senior executive political background were set according to whether they had served or were serving as national or provincial NPC deputies, CPPCC members, central, provincial or local municipal government administrative officials (department level or above); dummy variables for executive R&D background were set according to whether they had a working background in R&D positions).

First, ownership firms were divided into non-state-owned firms and state-owned firms (SOEs), both of which face different environments that affect the strategic decisions. SOEs face weaker external financing constraints and internal resource competition pressures while exhibiting strong political advantages such as easy access to scarce governmental resources and legitimacy recognition. However, SOEs are affected by the lack of R&D and inefficient innovation. Thus, further investigations on how the interaction between charitable giving and R&D innovation is characterized in different ownership firms is advocated.

Secondly, senior executives, as the absolute subjects of corporate governance, control and deploy resources in all aspects of the firm, undertake important missions such as corporate strategy formulation and decision implementation, and are therefore important players determining the business development of a firm [59]. From the social capital perspective, as the "insiders" of political organizations (Polbgd), senior executives have access to more information about the government, which can help firms effectively assess the possibility of receiving appropriate returns through charitable giving, thus reducing strategic risks and optimizing R&D innovation [4]. In terms of professional capital, senior executives with R&D backgrounds (RDbgd) are more responsive to the market demand for new technologies and focus on the firm's technological innovation. This suggests that they are likely to allocate resources in the direction of R&D innovation [60]. Having a professional technical background enables executives to accurately anticipate the technical difficulties of R&D projects and to rationalize resources to mitigate unforeseen technical uncertainties [61]. Overall, senior executive attributes significantly influence the synergistic relationship between charitable giving and R&D innovation [62].

Given that the synergy effect of charitable giving on R&D innovation is mainly prominent in mature firms, this paper further investigated this synergy based on mature firms, and the regression results are shown in Table 10. Data shown in columns (1) and (2) indicate that the synergy between charitable giving and R&D innovation is significantly stronger for non-SOEs (β = 11.877, *p* < 0.01) than for SOEs (β = 1.724, *p* < 0.01), which may be ascribed to the possibility that mature firms are less constrained by resources in making strategic decisions, and the synergy between them is significantly enhanced by the well-established management processes and internal governance structures of non-SOEs. In columns (3) and (4), the synergy between charitable giving and R&D innovation is significantly stronger for firms with executives with R&D backgrounds (β = 12.035, *p* < 0.01) than for firms with executives without R&D backgrounds (β = 4.94, *p* < 0.05). This is so because senior executives with an R&D background make decisions that significantly reduce the risk of R&D projects at the technical level and develop the skills to transform the firm's professional capital, which facilitates the donation behavior. The regression results in columns (5) and (6) indicate that the synergy between charitable giving and R&D in-novation is stronger for firms without political affiliations (β = 10.254, *p* < 0.01). The reason lies in the short-sighted behavior of senior executives. To achieve career development, executives with political backgrounds consider factors such as reputation and social status, are eager to enhance

their political recognition by means of charitable giving, and are hence likely to cause excessive giving behavior which may crowd out R&D resources, and the synergy between them is relatively weak.

**Table 10.** Further analysis of the synergy effect between charitable giving and R&D Innovation.

| Variables | Tobit Model | | | | | |
|---|---|---|---|---|---|---|
| | R&D1 | | | | | |
| | (1) | (2) | (3) | (4) | (5) | (6) |
| | SOEs | Non-SOEs | RDbgd | Non-RDbgd | Polbgd | Non-Polbgd |
| Don1 | 1.724 *** | 11.877 *** | 12.035 *** | 4.940 ** | 9.021 | 10.254 *** |
| | (3.007) | (4.294) | (7.662) | (2.393) | (1.696) | (3.210) |
| Control | Yes | Yes | Yes | Yes | Yes | Yes |
| In/Ye/Pr | Yes | Yes | Yes | Yes | Yes | Yes |
| Constant | −0.028 | −0.150 | −0.111 | −0.379*** | −0.645 | −0.060 |
| | (−0.895) | (−1.332) | (−1.361) | (−2.898) | (−1.578) | (−0.985) |
| Pseudo R2 | −0.595 | −0.524 | −0.754 | −0.464 | −1.375 | −0.360 |
| N | 152 | 213 | 99 | 266 | 30 | 335 |

Note: Robust *t*-statistics in brackets; ** $p < 0.5$, *** $p < 0.01$.

## 6. Discussion

### 6.1. Implications for Theory and Research

The theoretical implications of this paper are as follows. Although the prevailing research on the relationship between charitable giving and R&D innovation has resulted in the formation of a relatively complete theoretical foundation, it has not reached a consistent conclusion. This is because the existing studies have only examined the resource level of firms from a "one-dimensional" static perspective, mainly involving government subsidies, financial slack, and other types of resources, and they all explore the impact of resource level waxing and waning on the relationship between them within a single time dimension, ignoring the longitudinal examination of the resource capacity of firms at different life-cycle stages. Therefore, their conclusions are different from those of the present study. This study introduces the life-cycle theory of firms and comprehensively investigates the impact of the difference in resource levels of firms at different life-cycle stages on the relationship between them in a "two-dimensional" perspective. This approach avoids the limitations in the existing literature and provides a new perspective to explain the uncertainty of the empirical results between charitable giving and R&D innovation.

### 6.2. Managerial Implications

In summary, this article offers three main managerial implications:

First: Under the current context of the intensification of the game of great powers, high-tech enterprises should understand the high-value interaction between corporate charitable behavior and R&D innovation to solidify their innovation chain. R&D innovation should not just rely on the internal strength of the firm, but there is a need to involve more external stakeholders in the firm's innovation activities by means of charitable giving. This pools key scarce resources for R&D innovation and enhances the strength of the firm to cope with the external environmental impact.

Second: High-tech enterprises should further optimize resource allocation on the basis of perfecting strategic layout and combining with their own development reality to maximize the value of firm resources. Although, in general, the occurrence of charitable behavior contributes to higher levels of R&D innovation, its utility is affected by differences in the life cycle of firms. For mature and declining firms, with past business accumulation and strong financial strength, charitable giving achieves the public social responsibility needs while securing key scarce resources and support from external stakeholders, which

enhances organizational resilience and improves R&D innovation. For growth-stage firms, the existing financial constraints and the long return period of charitable giving make it difficult to bring in the social capital needed for R&D innovation in the short term. Thus, firms should avoid using charitable giving to fulfill their social responsibility to reduce the crowding out of R&D resources.

Third: Considering the influence of firm ownership form and senior executive attributes on the synergy relationship between charitable giving and R&D innovation, non-SOEs whose senior executives have R&D background should more actively engage in public welfare, both internally and externally, to build a good firm image and enhance its visibility externally. This will attract external advantageous resources to expand the firm's strength internally and promote R&D innovation development.

Fourth: The differentiated utility between charitable giving and R&D innovation over different life cycles will form a basis for attracting government subsidies in terms of research funding to high-tech enterprises. Such government subsidies are aimed at stimulating core technological changes tilted according to the life-cycle stage of the firm. Compared with mature firms, high-tech enterprises in the growth stage face serious financial constraints, and their strategic choices are largely limited by resources. In addition, firms with relatively weak capital portfolios are unable to obtain key social capital through charitable giving that can be fed into R&D innovation, and they may choose to avoid giving based on the consideration of the initial capital investment and return cycle. This makes it difficult for them to form a comprehensive strategic layout. In such cases, the financial assistance from the government can relieve financial pressure on firms while giving them more options and thinking space. It also projects key signals to the outside world to help them absorb the necessary political capital and social investment, relieving the pressure on their R&D innovation from the outside in.

*6.3. Limitations and Future Research Directions*

Given the significant impact of external environmental changes on R&D innovation by high-tech enterprises in recent years, this paper selected high-tech enterprises listed on Chinese A-shares from 2015 to 2020 as the research sample. However, since high-tech enterprises in China are in the initial stage of development, the samples are mostly concentrated in the growth and mature stages, and the number of samples in the decline stage is limited. In subsequent studies, there is a need to explore the utility of charitable giving and R&D innovation at different stages using a large sample by expanding the time interval and sample selection range. Secondly, this paper only examines the impact of one resource modality, charitable giving, on R&D innovation under the strategic resource perspective. Therefore, subsequent studies should investigate the impact of other resource modalities, such as financial slack and government subsidies, on R&D innovation under different life-cycle perspectives.

## 7. Conclusions

Based on the dynamic perspective of the corporate life cycle, this paper explored the strategic interaction between charitable giving and R&D innovation in high-tech enterprises in different life cycle stages. The research showed that there was a significant synergy between charitable giving and R&D innovation, especially for mature firms with abundant capital. Donation can reflect the significance of constructing social capital, bring policy dividends and scarce resources for R&D innovation, and protect the R&D process from the impact of the external environment. The synergy performance of declining firms was second, but it was not significant for growing firms. The reasons are that, first, the strategic decisions of growth-stage firms are inevitably affected by resource constraints. Second, the weak social network of growth firms leads to a long payback period for charitable giving, which cannot meet the demand of R&D innovation for social capital in the short term, but will aggravate the crowding out of R&D resources.

At the same time, this paper divided the context based on the form of firm ownership and senior executive attributes, analyzed the specific application of their synergy relationship, and expanded the research on the economic utility of donation behavior. The results showed that the synergy between charitable giving and R&D innovation was better in non-SOE firms. From the perspective of executives' attributes, it is easier for firms where executives have an R&D background to give play to the role of charitable donations in promoting R&D innovation. On the contrary, the political attributes of executives can exacerbate agency problems, leading to excessive donation, which was not conducive to R&D innovation. Finally, the regression findings remained significant after accounting for possible endogeneity and heteroskedasticity between charitable giving and R&D innovation.

**Author Contributions:** Conceptualization, Y.Z. and Z.L.; methodology, Y.Z.; software, Y.Z.; validation, Z.L. and Y.L.; formal analysis, Y.Z.; investigation, Y.Z.; resources, Y.Z.; data curation, Y.Z.; writing—original draft preparation, Y.Z.; writing—review and editing, Z.L. and Y.L.; visualization, Y.Z.; supervision, Z.L.; project administration, Z.L.; funding acquisition, Z.L. All authors have read and agreed to the published version of the manuscript.

**Funding:** This research received no external funding.

**Institutional Review Board Statement:** Not applicable.

**Informed Consent Statement:** Not applicable.

**Data Availability Statement:** The dataset generated and analyzed in this study is not publicly available. The dataset is available from the corresponding author on reasonable request.

**Conflicts of Interest:** The authors declare no conflict of interest.

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
