# Peer review of "Strategic Charitable Giving and R&D Innovation of High-Tech Enterprises: A Dynamic Perspective Based on the Corporate Life Cycle"

_sustainability, doi:10.3390/su142316180_

Round 1

Reviewer 1 Report

This study aims to test the interaction between charitable giving and R&D innovation and the differences in their utility exhibited over different life cycles. There are some comments that are recommended to be addressed prior to consideration of publication.

1.       It is recommended to add at the end of the introduction section a paragraph in order to set out the content of the rest of the paper.

2.       Lines 100-102: Please answer how is that supported? Is it an authors’ hypothesis? Or is it supported by other studies? Please clarify and add the relevant sources. Please do the same for 127-134, 202-218, 225-232, 249-264, 271-277,  282-290 lines.

3.       Please follow the references rules (Dickinson – Lines 153, 158). Re-examine carefully the whole range of the text in order to make the corresponding corrections.

4.       Please add source for the Database of China Stock Market & Accounting Research Database.

5.       Lines 301-312: Please justify with the help of evidence from literature why you selected the 3-year validation of enterprises.

6.       I recommend you not using the First Person Singular or Plural. Usually, the third one is appropriate for a research paper. Example: Present study’s authors merged… (Not: We propose, we need, we merge e.t.c.).

7.       Please justify with the help of evidence from literature why you have selected grouping (Lines: 322-343) and control variables (Lines: 345-352).

8.       Please justify with the help of evidence from literature how statistical Techniques are supported (353-368). Same for Endogeneity test (Lines: 438-450).

9.       English should be revised. Please re-examine the whole range of the text. 

Author Response

Thank you very much for your review on 9th November 2022 in which you give the review report on our paper with the manuscript ID sustainability-2046827. We have carefully revised the manuscript according to your suggestion.

Our responses to several comments are listed below:

Comment 1:

It is recommended to add at the end of the introduction section a paragraph in order to set out the content of the rest of the paper.

Reply:

We have added a paragraph at the end of the introduction section to describe the structure of the rest of the paper, please see the revised manuscript for details.

Comment 2:

Lines 100-102: Please answer how is that supported? Is it an authors’ hypothesis? Or is it supported by other studies? Please clarify and add the relevant sources. Please do the same for 127-134, 202-218, 225-232, 249-264, 271-277, 282-290 lines.

Reply:

I have thoroughly examined and reviewed the content of the paper to which you refer (100-102, 127-134, 202-218, 225-232, 249-264, 271-277, 282-290), and have fully quoted the views and contents from other studies, an unquoted part is considered to be the view of the paper.

Comment 3:

Please follow the references rules (Dickinson – Lines 153, 158). Re-examine carefully the whole range of the text in order to make the corresponding corrections.

Reply:

I have revised this question and carefully examined all the references in the whole range of the text.

Comment 4:

Please add source for the Database of China Stock Market & Accounting Research Database.

Reply:

China Stock Market & Accounting Research Database (CSMAR) is a research-oriented accurate database in the economic and financial field developed by Shenzhen Xishima Data Technology Co., Ltd. from the academic research needs, drawing lessons from CRSP, COMPUSTAT, TAQ, THOMSON and other authoritative database professional standards, and combining the actual situation of China. It is also a data source for more than 500 universities and financial institutions in China to conduct empirical analysis on securities, economics and finance. I have added the above in the appropriate places in our paper.

Comment 5:

Lines 301-312: Please justify with the help of evidence from literature why you selected the 3-year validation of enterprises.

Reply:

According to Bi et al., in China, the validity period of the accreditation of high-tech enterprises is generally three years, and it is necessary to re-accredit after the expiration of the period. Therefore, this paper obtained the accreditation dates of high-tech enterprises from 2015 to 2020, taking the data within three years after their accreditation as the sample period, and based on this, obtained an initial data with 14,684 samples.

Comment 6:

I recommend you not using the First Person Singular or Plural. Usually, the third one is appropriate for a research paper. Example: Present study’s authors merged… (Not: We propose, we need, we merge e.t.c.).

Reply:

I checked the whole content repeatedly and replaced all the First Person Singular or Plural with "This study, This paper" , and marked the changes in the text.

Comment 7:

Please justify with the help of evidence from literature why you have selected grouping (Lines: 322-343) and control variables (Lines: 345-352).

Reply:

For the part of selected grouping and control variables, I have quoted the literature I have used for reference, and the unquoted part is the viewpoint of the paper.

Comment 8:

Please justify with the help of evidence from literature how statistical Techniques are supported (353-368). Same for Endogeneity test (Lines: 438-450).

Reply:

For the part of statistical techniques and Endogeneity test, I have quoted the literature I have used for reference, and marked the changes in the text.

Comment 9:

English should be revised. Please re-examine the whole range of the text.

Reply:

With the help of a native English speaker, the grammar, spelling, punctuation and phrasing of the whole text has been checked.

Details of other revisions made at the suggestion of other referees.

  1. The penultimate paragraph of the introduction has been appropriately revised, inappropriate expressions and redundant parts were deleted.
  2. A framework is added to show the research model of the article more intuitively.
  3. We added a description of quantitative research.
  4. Some references have been updated.
  5. In order to make the table more concise, the shorthand symbols of variables are uniformly adjusted.
  6. In order to make the research results easier to understand, we revised the description of the research conclusion.

We have marked all the revised contents in the paper. Thank you again for your review of our paper.

Reviewer 2 Report

First, I appreciate the opportunity to be a reviewer for such an article. The authors draw attention to the interaction between charities donations and innovation in research and development and the differences in their usefulness manifested over different life cycles. The results show that there is synergy between philanthropic giving and R&D innovation, philanthropic giving can significantly enhance innovation in research and development, but their utility is affected by changes in the life cycle of firms.

After reading the research and analysis of the authors of the article, I would make the following comments and suggestions:

I. Originality:

The paper's research topic is topical as it covers methods of charitable giving and innovation in research and development. They are aimed at high-tech enterprises for improvement and utility. It is the study of the influence of phenomena that contributes to the originality of the article.

II. Literature review:

The use of more than 50 different sources of literature shows the variety of ideas with which the authors have become familiar and on the basis of which they have expanded their field of knowledge. 

III. Methodology:

The methodology used is comprehensive and the data prove the dependencies between the investigated indicators. This is exactly why I think that the authors have correctly put forward 4 hypotheses that have subsequently been accepted or rejected. This was done on the basis of the objective results already obtained. 

IV. Results and discussion:

The results are well presented and the discussion outlines the reasons for them.

V. Quality of communication:

The article is well written and easy to follow. However, the research and data are presented in such a way that it would be difficult for an ordinary eye to understand the results of the research and analysis. 

In conclusion, I think that the presence of hypotheses and, accordingly, arriving at their answers, which exist in the article, increase the scientific and practical qualities of the development. Also, the specific data and proven claims enrich the current literature. The clearly expressed results and drawn conclusions prove the completeness and thoroughness of the article.

Author Response

Thank you very much for your review on 21st November 2022 in which you give the review report on our paper with the manuscript ID sustainability-2046827. I read your revised report carefully and thank you for your affirmation of the quality of our paper.

As for your question that " the research and data are presented in such a way that it would be difficult for an ordinary eye to understand the results of the research and analysis", The content of our paper has been revised as follows:

  1. In order to make the table more concise, the shorthand symbols of variables are uniformly adjusted.
  2. Some table structures are adjusted in order to achieve clearer expression.
  3. In order to make the research results easier to understand, we revised the description of the research conclusion.
  4. With the help of a native English speaker, the grammar, spelling, punctuation and phrasing of the whole text has been checked.

Details of other revisions made at the suggestion of other referees.

  1. The penultimate paragraph of the introduction has been appropriately revised, inappropriate expressions and redundant parts were deleted.
  2. We added a paragraph at the end of the introduction section to describe the structure of the rest of the paper.
  3. We quoted the views and contents from other studies fully.
  4. A framework is added to show the research model of the article more intuitively.
  5. We revised the citation format of some references and carefully examined all the references in the whole range of the paper.
  6. We added source for the Database of China Stock Market & Accounting Research Database.
  7. We added relevant references to justify why we selected the 3-year validation of enterprises.
  8. We added a description of quantitative research.
  9. We checked the whole content repeatedly and replaced all the First Person Singular or Plural with "This study, This paper", and marked the changes in the text.
  10. We added relevant references to justify why we selected grouping and control variables.
  11. We added relevant references to justify how statistical Techniques and Endogeneity test were supported.
  12. Some references have been updated.

We have marked all the revised contents in the paper. Thank you again for your review of our paper. 

Reviewer 3 Report

1-the introduction quite long and should not put summery in introduction 

2- I recommend the draw the framework 

3-Did you use qualitative or quantitative research? justify

4- analysis should be more clear and result 

5- some of references need to be update 

Author Response

Thank you very much for your review on 20th November 2022 in which you give the review report on our paper with the manuscript ID sustainability-2046827. We have carefully revised the manuscript according to your suggestion.

Our responses to several comments are listed below:

Comment 1:

The introduction quite long and should not put summery in introduction.

Reply:

The penultimate paragraph of the introduction has been appropriately revised, inappropriate expressions and redundant parts were deleted, and marked the changes in the text.

Comment 2:

I recommend the draw the framework.

Reply:

A framework is added to show the research model of the article more intuitively.

Comment 3:

Did you use qualitative or quantitative research? Justify.

Reply:

We added a description of quantitative research in line 409.

Comment 4:

Analysis should be more clear and result.

Reply:

In order to make the table more concise, the shorthand symbols of variables are uniformly adjusted and some table structures are adjusted in order to achieve clearer expression. And also, we revised the description of the research conclusion to make the research results easier to understand.

With the help of a native English speaker, the grammar, spelling, punctuation and phrasing of the whole text has been checked.

Comment 5:

Some of references need to be update

Reply:

Some references have been updated, including 7, 8, 9, 14, 41, 42, 46. And some references have been added.

Details of other revisions made at the suggestion of other referees.

  1. We added a paragraph at the end of the introduction section to describe the structure of the rest of the paper.
  2. We quoted the views and contents from other studies fully.
  3. We revised the citation format of some references and carefully examined all the references in the whole range of the paper.
  4. We added source for the Database of China Stock Market & Accounting Research Database.
  5. We added relevant references to justify why we selected the 3-year validation of enterprises.
  6. We checked the whole content repeatedly and replaced all the First Person Singular or Plural with "This study, This paper", and marked the changes in the text.
  7. We added relevant references to justify why we selected grouping and control variables.
  8. We added relevant references to justify how statistical Techniques and Endogeneity test were supported.

We have marked all the revised contents in the paper. Thank you again for your review of our paper.

Round 2

Reviewer 1 Report

no comments

Reviewer 3 Report

well done